# Nutritional status and TB treatment outcomes in Addis Ababa, Ethiopia: An ambi-directional cohort study

Zekariyas Sahile[1]*, Robel Tezera[2], Damen Haile Mariam[3], Jeffrey Collins[4], Jemal Haider Ali[3]

1 Department of Public Health, College of Medicine and Health Science, Ambo University, Ambo, Ethiopia, 2 Department of Radiology, School of Medicine, Addis Ababa University, Addis Ababa, Ethiopia, 3 School of Public Health, College of Health Science, Addis Ababa University, Addis Ababa, Ethiopia, 4 School of Medicine, Emory University, Georgia, Atlanta, United States of America

* zekitiru@gmail.com

## Abstract

### Background

Remaining underweight during Tuberculosis (TB) treatment is associated with a higher risk of unsuccessful TB treatment outcomes and relapse. Previous studies conducted in Ethiopia found that bodyweight not adjusted for height at the start of treatment is associated with poor treatment outcomes. However, the association of body mass index (BMI) and weight change during treatment with treatment outcomes has not been studied. We aimed to investigate the association of BMI at the time of diagnosis and after two months of treatment and TB treatment outcomes.

### Methods

Using an ambi-directional cohort study design (retrospective and prospective), a total of 456 participants were enrolled among 30 randomly selected public health centers residing within six sub-cities of Addis Ababa, Ethiopia. Data were collected using medical chart abstraction and face to face interviews. We compared TB treatment outcomes in persons with a body mass index (BMI) <18.5kg/m$^2$ (underweight) versus persons with BMI $\geq$18.5kg/m$^2$ (normal or overweight) at treatment initiation and after two months of treatment. Treatment was classified as successful in persons who were free of symptoms and had a negative sputum smear for acid-fast bacilli at the end of the 6-month treatment course. We analysed outcomes using univariable and multivariable logistic regression with 95% CI and p value< 0.05.

### Results

Of enrolled study participants, 184 (40.4%) were underweight and 272 (59.6%) were normal or overweight. Body mass index (BMI $\geq$18.5kg/m$^2$) at the start and second month of treatment were independent predictors for successful treatment outcome (AOR = 2.15; 95% CI: 1.05, 4.39) and (AOR = 3.55; 95% CI: 1.29, 9.73), respectively. The probability of treatment

**Data Availability Statement:** The minimal data set is available in the paper. Additional data is available upon reasonable request due to identifiable information. A formal request could be made to the

Addis Ababa Public Health Research and
Emergency Management Core process ethics
committee for additional data.

**Funding:** This research work was funded by Addis
Ababa University thematic research project fund
and supported in part by the NIH/Fogarty
International Center, Global Infectious Disease 341
grant D43TW009127.

**Competing interests:** The authors declare they
have no conflict of interest.

success among patients with BMI$\geq$18.5kg/m$^2$ at the start and second month of treatment was 92.9% and 97.1%, respectively versus 86.5% and 91.7% in patients with BMI<18.5kg/m$^2$. Bodyweight not adjusted for height and change in the bodyweight after the second and sixth months of treatment were not significantly associated with treatment success.

## Conclusion

In persons treated for TB disease, being underweight at baseline and after two months of treatment was a predictor for unsuccessful treatment outcomes. Nutritional assessment, counselling, and management are important components of TB treatment programs with the potential to improve treatment outcomes.

## Background

Tuberculosis (TB) is the leading cause of death from an infectious disease worldwide. In Ethiopia in 2018, the total TB incidence rate was 151 per 100, 000 population, and the TB mortality rate was 22 per 100,000 population in persons without HIV [1]. Despite significant improvements in the TB treatment outcomes following the introduction of DOTS services in Ethiopia, many regions continue to report a large number of unsuccessful treatment outcomes: in the Amhara region 39.9% [2], in the Tigray region 10.8% [3], in the South region 14.8% [4] and in Addis Ababa 9.02% [5]. Therefore, there exists a critical need to identify the causes and predictors of unsuccessful TB treatment outcomes in Ethiopia.

Undernutrition is associated with an increased risk of mortality and relapse in persons with TB disease [6–8]. The severity of lung disease in adults with pulmonary TB (PTB) is also associated with the extent of malnutrition, as measured by body mass index (BMI) and body composition [9]. Undernourished patients have been shown to have poor bioavailability of key drugs like rifampicin which can contribute to treatment failure and development of multidrug resistance [10], and being undernourished is a risk factor for hepatotoxicity on TB treatment [11–13]. Thus, in Ethiopia, where 40–70% of patients treated for TB disease are underweight at diagnosis [14–19], undernutrition may be a major contributor to unsuccessful treatment outcomes.

Although several studies demonstrate an association between nutritional status and TB treatment outcomes, it is not yet well studied in the Ethiopian context. Existing studies in Ethiopia show that low body weight not adjusted for height at the start of treatment is associated with unsuccessful TB treatment outcomes [2,20–22]. However, no studies in Ethiopia to date have examined the association between TB treatment outcomes and the more accurate measure of nutritional status, BMI. Additionally, it remains unclear whether a change in BMI after initiation of anti-TB therapy is independently associated with successful treatment outcomes beyond baseline nutritional status. We sought to determine whether being underweight, defined as BMI<18.5kg/m$^2$, at baseline and after 2 months of treatment are independently associated with successful treatment outcomes in Ethiopian patients treated for TB disease.

## Methods

### Study design and setting

We employed an ambi-directional cohort study design (retrospective and prospective) to assess the association between nutritional status and TB treatment outcomes. Weight and

height were taken from retrospective registered data of patients diagnosed with TB disease and the exposure status was classified as underweight (BMI<18.5kg/m$^2$) and normal or overweight (BMI $\geq$18.5kg/m$^2$).

The study was conducted in Addis Ababa, the capital city of the Federal Democratic Republic of Ethiopia. Addis Ababa is divided into 10 administrative sub-cities. Patents who remained alive after 3 months of TB treatment were interviewed and followed prospectively for a maximum of 3 months to the end of the 6 month treatment period. Data collection was conducted from the first week of September to the end week of November 2019.

## Participants

Participants diagnosed with TB disease were enrolled from 30 randomly selected health facilities residing within six sub-cities of Addis Ababa, Ethiopia. Participants with BMI<18.5kg/m$^2$ were classified as underweight and participants with BMI $\geq$18.5kg/m$^2$ were classified as normal or overweight. All participants were treated, according to the national guideline for TB treatment [23], with two months of rifampicin, isoniazid, pyrazinamide and ethambutol followed by four months of rifampicin and isoniazid.

## Eligibility

Persons diagnosed with TB disease who initiated anti-TB treatment for at least 3 months and aged 18 years and older were eligible for this study. Exclusion criteria were: (1) the care of the patient was transferred- in or out of the health facility, (2) the patient was treated for longer than six months, (3) the patient was diagnosed with MDR-TB, or (4) the patient was pregnant or lactating.

## Variables

TB patients who had BMI <18.5kg/m$^2$ were classified as underweight and TB patients who had BMI $\geq$18.5kg/m$^2$ at the start of the treatment were classified as normal or overweight. We calculated absolute change in weight in kilograms relative to baseline at the second and sixth months of treatment.

Treatment was classified as successful if the patient had a negative acid-fast bacilli (AFB) sputum smear at the end of treatment and at least one previous test or completed treatment with resolution of symptoms. Treatment was considered unsuccessful in a patient with a persistently positive AFB sputum smear at 5 months or later during TB treatment, or died or defaulted from treatment.

## Sample size

The sample size was calculated using Epi Info statistical software using a two population proportion sample size calculation for cohort study with a confidence level of 95% ($\alpha$ = 0.05) and statistical power of 80% ($\beta$ = 0.20). Using treatment outcomes from a prior study conducted in Myanmar and Zimbabwe [24], we estimated unsuccessful treatment outcomes (treatment failure, default, death) would occur in 34.1% of underweight patients (P1 = 0.341) and 18% of the normal or overweight patients (P2 = 0.182). We estimated a 3:2 ratio of normal or overweight to underweight patients. Based on these assumptions the initial sample size was calculated to be 270 (108 in the underweight and 162 in the normal or overweight groups). We further estimated an average cluster size of 10 and an inter-correlation coefficient (ICC) of 0.05 yielding a design effect of 1.5. Multiplying the initial sample size by a design effect of 1.5 yielded 405 (108x1.5 = 162 in the underweight and 162x1.5 = 243 in the normal or overweight groups).

Considering a 10% non-response rate and a 5% loss to follow-up, the final sample size was 466 (186 in the underweight and 280 in the normal or overweight groups).

## Sampling

A total of 96 functional public health centers were providing DOTS service in 10 sub-cities of Addis Ababa. A total of 30 functional public health centers (5 health centers from each sub-city) were randomly selected from six sub-cities (Nifas-Silk Lafto, Kolfe Keranio, Akaki Quality, Kirkos, Bole, and Lideta). The sample size was allocated based on the probability proportional to size (PPS) of the population in each facility; health centers with large amounts of patients were allocated large sample sizes for both the underweight and normal or overweight patients.

## Data collection methods

The questionnaire for this study was prepared using WHO guidelines [25, 26] and similar related studies [27–32]. The questionnaire has six sections that assess the demographic characteristics, clinical characteristics, nutritional status, nutritional counselling and support, treatment outcomes, and AFB sputum smear results for study participants. Data were collected through face-to-face interviews with patients and medical document review by trained health professionals. The investigators were not involved in the selection of study participants or the data collection process. Data collection included reviewing TB registration books to identify eligible participants and classified the exposure status by BMI as BMI<18.5kg/m$^2$ or BMI≥18.5kg/m$^2$ [26]. To assess the weight change, patients' weight was recorded at the start of treatment, in the second month, and at the end of six months of treatment from TB registration books. Information on socio-demographics, and availability of nutritional counselling and support were collected by face to face interviews. Nutritional counselling was defined as receipt of any nutrition related information or guidance by clinic staff. Individual nutrition support programs varied by study site. Participants' AFB sputum smear results were collected from the review of the TB registration book at two, five, and end of six months of treatment. Treatment outcomes were assessed based on the WHO classification of treatment outcomes [25] by reviewing TB registration books.

## Statistical analysis

The data were edited, checked for completeness, coded, and entered in Epi Info statistical software version 7.2 and exported to STATA software version 13 for further analysis. BMI was calculated from weight and height and categorized according to WHO standards [26]. The weight change was calculated by the absolute change in weight in kilograms. Associations between underweight status and demographic, clinical, nutritional and food support variables were assessed using a Chi-square test. Univariable and multivariable logistic regression were used to assess associations between underweight status and treatment outcomes adjusted to control for the effect of confounding variables. An association with p-value < 0.05 and 95%CI were considered statistically significant. The predicted probability of successful treatment outcomes was calculated by BMI and covariates including age and gender using a multivariable logistic regression model. Analyses were not corrected for multiple comparisons. Missing and loss to follow up data were not included in the analysis.

## Ethical consideration

An ethical clearance approval letter was obtained from the Addis Ababa Public Health Research and Emergency Management Core process ethics committee. The study participants

provided written informed consent after receiving comprehensive information about the study objective, benefit, and any anticipated risk. The participants had the right to withdraw from the study at any time during the study period for any reason. Only the principal investigator had access to personal identifiers of study participants.

## Results

### Demographic characteristics

A total of 184 underweight patients (BMI<18.5 kg/m$^2$) and 272 normal or overweight patients (BMI≥18.5 kg/m$^2$) were included in the final analysis of the study (Fig 1). Of enrolled participants, 97.85% (98.92% in underweight, and 97.14% in normal or overweight patients) completed the study. The median age of the underweight and normal or overweight patients was 29 and 30 years, respectively. One hundred thirteen (61.41%) underweight patients and 138 (50.29%) normal or overweight patients were male. Ninety-two (51.69%) underweight patients were single whereas 139 (52.06%) normal or overweight patients were married (Table 1).

### Clinical characteristics

Overall, 207(45.39%) patients had smear-positive PTB; 110(59.78%) underweight and 97 (35.66%) normal or overweight patients. Most of the patients (95.39%) were newly diagnosed with TB; 167(90.76%) in the underweight group and 268(98.53%) in the normal or overweight group. A higher proportion of underweight patients (23.3%) had TB/HIV co-infection versus normal or overweight patients (13.60%). Regarding non-communicable diseases,

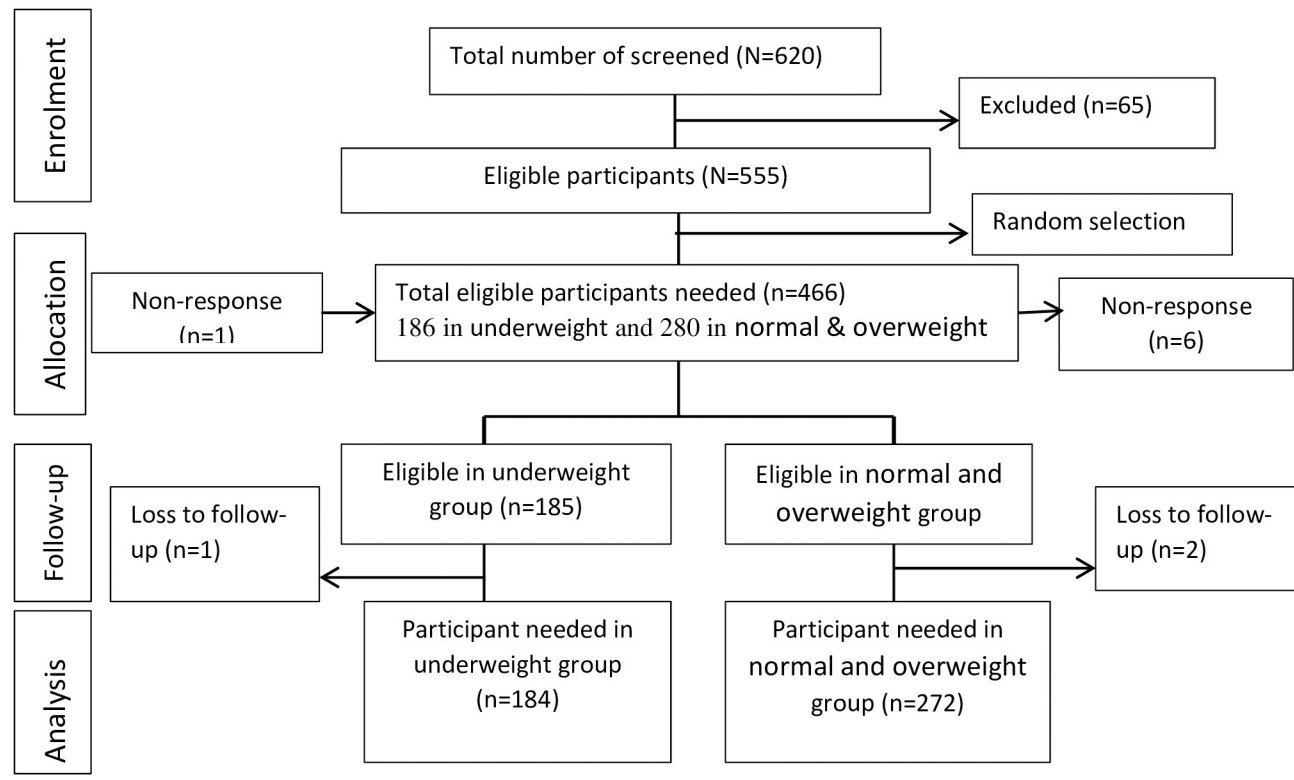

**Fig 1. Schematic presentation of the sampling process among adult TB patients at Addis Ababa, Ethiopia, 2019.**

**Table 1. Demographic characteristics of adult TB patients in public health centers of Addis Ababa, Ethiopia, 2019.**

| Variables | Underweight (BMI<18.5kg/m$^2$) | Normal or overweight (BMI≥18.5kg/m$^2$) | Both groups | P-value |
|---|---|---|---|---|
| **Sex** | **n1 = 184** | **n2 = 272** | **N = 456** | |
| Male | 113(61.41%) | 138(50.29%) | 251(55.045) | 0.025 |
| Female | 71(38.59%) | 134(49.26%) | 205(44.96%) | |
| **Marital status** | **n1 = 178** | **n2 = 267** | **N = 445** | |
| Single | 92(51.69%) | 102(38.20%) | 194(43.60%) | 0.031 |
| Married | 69(38.76%) | 139(52.06%) | 208(46.74%) | |
| Divorced | 12(6.74%) | 16(5.99%) | 28(6.295) | |
| Widowed | 5(2.81%) | 10(3.75%) | 15(3.37%) | |
| **Education** | **n1 = 172** | **n2 = 263** | **N = 435** | |
| No formal education | 23(13.37%) | 41(15.59%) | 64(14.71%) | <0.00001 |
| 1–8 grade | 50(29.07%) | 61(23.19%) | 111(25.52%) | |
| 9–10 grade | 59(34.30%) | 43(16.35%) | 102(23.45%) | |
| 11–12 grade | 13(7.56%) | 49(18.63%) | 62(14.25%) | |
| Above 12 grade | 27(15.70%) | 69(26.24%) | 96(22.07%) | |
| **Occupation** | **n1 = 170** | **n2 = 262** | **N = 432** | |
| Governmental | 15(8.82%) | 49(18.70%) | 64(14.81%) | 0.020 |
| Merchant | 26(15.29%) | 48(18.32%) | 74(17.13%) | |
| Private worker | 23(13.53%) | 38(14.50%) | 61(14.12%) | |
| Daily labor | 49(28.82%) | 63(24.05%) | 112(25.93%) | |
| Driver | 4(2.35%) | 6(2.29%) | 10(2.31%) | |
| Student | 21(12.4%) | 16(6.11%) | 37(8.56%) | |
| Housewife | 21(12.35%) | 35(13.36%) | 56(12.96%) | |
| Others* | 11(6.47%) | 7(2.68%) | 18(4.17%) | |

*Pensioner, House servant, Guard.

9(4.97%) underweight patients, and 14(5.15%) normal or overweight patients had hypertension (Table 2).

## Nutritional support

One hundred thirty (77.8%) underweight patients and 203 (78.6%) normal or overweight patients received nutritional counselling. However, nearly one in four underweight patients (22.16%) and 21.32% of normal or overweight patients did not receive nutritional counselling. Additionally, 83.83% (140/167) of underweight patients and 92.64% (239/258) of normal or overweight patients did not receive food support from any sources (Table 2). We did not find any association between receipt of nutritional counselling and support and change in body weight and BMI after 2 months of treatment (S1 Table).

## Treatment outcomes

The overall proportion of unsuccessful TB treatment outcomes was 9.87% (45/456). The proportion of unsuccessful treatment outcomes among underweight and normal or overweight patients was 14.7%(27/184) and 6.6%(18/272), respectively. The death rate was higher in the underweight patients (9.24%; 17/184) than in normal or overweight patients (5.15%; 14/272). In those who died, the median time to death was 42 days: 59 days underweight patients and 36.5 days in normal or overweight patients. Among the 106 PTB patients in the underweight group completing six months of TB treatment, 8 (7.55%) continued to have a positive AFB

**Table 2. Clinical characteristics and nutritional support of adult TB patients in public health centers of Addis Ababa, Ethiopia, 2019.**

| Variables | Underweight (BMI<18.5kg/m²) | Normal or overweight (BMI≥18.5kg/m²) | Both groups | P-value |
|---|---|---|---|---|
| **Type of TB and AFB sputum smear positivity** | **n1 = 184** | **n2 = 272** | **N = 456** | |
| Smear positive pulmonary TB | 110(59.78%) | 97(35.66%) | 207(45.39%) | <0.0001 |
| Smear negative pulmonary TB | 49(26.63%) | 74(27.21%) | 123(26.97%) | |
| Extra pulmonary TB | 25(13.59%) | 101(37.13%) | 126(27.63%) | |
| **History of TB** | **n1 = 184** | **n2 = 272** | **N = 456** | |
| New case | 167(90.76%) | 268(98.53%) | 435(95.39%) | <0.0001 |
| Relapsed | 15(8.15%) | 4(1.47%) | 19(4.17%) | |
| Treatment failure * | 2(1.09%) | 0 | 2(0.44%) | |
| **TB-HIV co-infected** | 44(23.3%) | 37(13.60%) | 81(17.76%) | 0.0047 |
| **Non-communicable disease** | **n1 = 181** | **n2 = 276** | **N = 453** | |
| Hypertension | 9(4.97%) | 14(5.15%) | 23(5.08%) | 0.2840 |
| Diabetes | 2(1.10%) | 8(2.90%) | 10(2.21%) | |
| Hepatitis B Infection | 0 | 2(0.74%) | 2(0.44%) | NA |
| **Nutritional counselling** | **n1 = 167** | **n2 = 258** | **N = 425** | |
| Yes | 130(77.84%) | 203(78.68%) | 333(78.35%) | 0.8376 |
| No | 37(22.16) | 55(21.32%) | 92(21.65%) | |
| **Food support** | **n1 = 167** | **n2 = 258** | **N = 452** | |
| Yes | 27(16.17%) | 19(7.36%) | 46(10.82%) | 0.0043 |
| No | 140(83.83%) | 239(92.64%) | 379(89.18%) | |

*Treatment failure merged with relapsed cases for chi-square calculation.

NA-Not applicable.

sputum smear, while all 90 smear-positive PTB patients in the normal or overweight group had a negative AFB sputum smear at the end of treatment.

## Univariable and multivariable analysis of BMI and weight change association with treatment outcomes

In the univariable analysis, being normal or overweight at the start and second month of treatment was significantly associated with successful TB treatment outcomes. The odds of successful treatment were 2.42 times greater in normal or overweight patients at baseline versus those that were underweight (95%CI 1.29, 4.55). Similarly, odds of successful treatment were 3.55 times higher in patients who were normal or overweight after 2 months of treatment versus those who were underweight at that time (95%CI 1.43, 8.78). After adjusting for sex, age, TB/HIV co-infection, type of TB and AFB sputum smear positivity, being normal or overweight at the start of treatment (AOR = 2.15; 95% CI: 1.05, 4.39) and the second month of treatment (AOR = 3.55; 95% CI: 1.29, 9.73) remained significantly associated with successful treatment outcomes. However, in univariable analysis, weight not adjusted for height at the start, second, and six months of treatment and weight change at the second-and six month time points relative to baseline were not significantly associated with TB treatment outcomes (Table 3, S2 and S3 Tables).

After adjusted for sex, age, TB/HIV co-infection, type of TB and AFB sputum smear positivity, the predicted probability of TB treatment success in normal or overweight patients was 92.9%, whereas this predicted probability of treatment success in the underweight patients was 86.5% (Fig 2). Findings were similar in the second month of treatment; the predicted

**Table 3. Univariable and multivariable analysis of BMI and weight change association with treatment outcomes among adult TB patients in the public health center of Addis Ababa, Ethiopia, 2019.**

| Explanatory Variables | Successful treatment outcomes | Unsuccessful treatment outcomes | UAOR | 95% CI | AOR | 95% CI |
|---|---|---|---|---|---|---|
| BMI at the start of treatment | | | | | | |
| BMI$\geq$18.5 kg/m$^2$ | 254 | 18 | 2.42 | 1.29, 4.55 | 2.15 | 1.05, 4.39 |
| BMI<18.5 kg/m$^2$ * | 157 | 27 | | | | |
| BMI after 2 months of treatment | | | | | | |
| BMI$\geq$18.5 kg/m$^2$ | 282 | 8 | 3.55 | 1.43, 8.78 | 3.55 | 1.29, 9.73 |
| BMI<18.5 kg/m$^{2*}$ | 129 | 13 | | | | |
| Bodyweight change | | | | | | |
| After 2 months of treatment mean (SD) | 1.83(3.63) | 1.38(2.71) | 1.03 | 0.92, 1.15 | | |
| After 6 months of treatment mean (SD) | 3.92(4.48) | 4(2.83) | 0.99 | 0.87, 1.13 | | |
| Bodyweight | | | | | | |
| At start of treatment mean (SD) | 52.61(10.07) | 50.80(12.10) | 1.01 | 0.98, 1.05 | | |
| After 2 months of treatment mean (SD) | 54.44(9.99) | 52.33(12.46) | 1.02 | 0.97, 1.07 | | |
| After 6 months of treatment mean (SD) | 56.53(9.72) | 55.67(11.78) | 1.00 | 0.95, 1.07 | | |

*referent, UAOR-Unadjusted odds ratio, AOR-Adjusted odds ratio, CI- Confidence interval, SD-Standard deviation.

probability of treatment successes in normal or overweight patients was 97.1% and the predicted probability of treatment successes in underweight patients was 91.7% (Fig 3).

After adjustment for age, TB/HIV co-infection, type of TB and AFB sputum smear positivity, the predicted probability of treatment success in normal or overweight male patients was 90.92%, while it was 96.20% in normal or overweight female patients. The predicted probability of treatment success in the underweight male patients was 81.61%, while it was 91.39% in the underweight female patients (S1 Fig). Similarly, the adjusted predicted probability of treatment successes in the second month of treatment in normal or overweight male patients was 95.27%, while it was 99.31% in normal or overweight female patients. This predicted probability of treatment success in the underweight male patients was 88.45%, while it was 98.12% in the underweight female patients (S2 Fig).

Similarly, after adjusted for sex, TB/HIV co-infection, type of TB and AFB sputum smear positivity, the predicted probability of treatment successes declined with increasing age in both

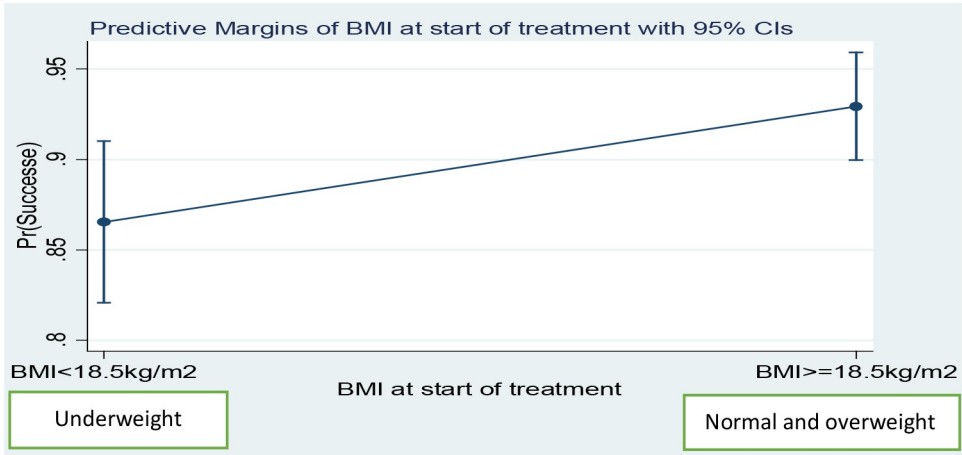

**Fig 2. Predicted probability of treatment successes by BMI at the start of treatment among adult TB patients in public health centers of Addis Ababa, Ethiopia, 2019.**

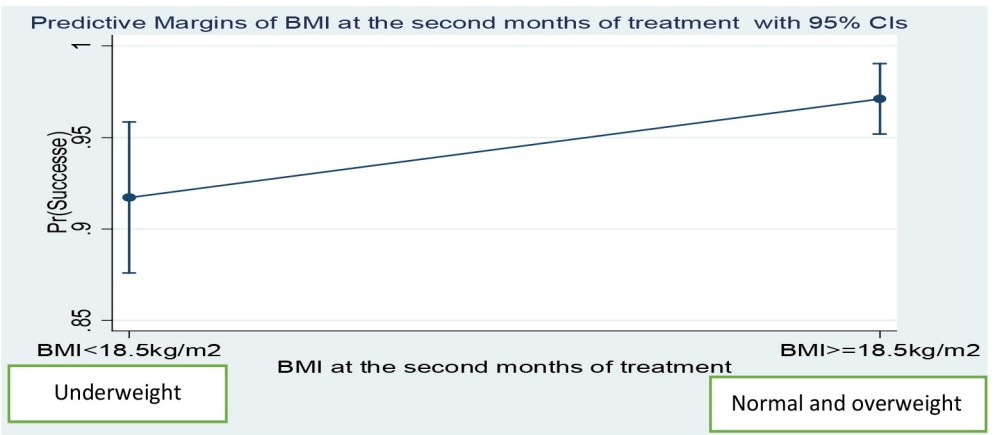

**Fig 3. Predicted probability of treatment successes by BMI at the start of treatment among adult TB patients in public health centers of Addis Ababa, Ethiopia, 2019.**

underweight and normal or overweight patients. For example, the predicted probability of treatment success among normal or overweight patients at 40 years of age was 92.6%, while it was 69.17% at 70 years of age. Similarly, the predicted probability of treatment success among underweight patients at age 40 was 84.86%, while it was 50.91% at age 70 (S3 Fig). The predicted probability of treatment success among patients that were underweight in the second month of treatment and at 40 years old was 90.76%, while it was 75.40% at age 70. Whereas the predicted probability treatment success among normal overweight in the second month of treatment and age 40 years was 96.96%, while it was 90.23% at age 70 years (S4 Fig).

## Discussion

In this study, we demonstrate that in patients treated for TB disease in Ethiopia, being normal or overweight at the start of treatment and after 2 months of treatment are independent predictors of successful treatment outcomes. The effect of being normal or overweight remained statistically significant after adjustment for sex, age, TB/HIV co-infection, type of TB and AFB sputum smear positivity. We also found an overall treatment success rate of 90.13%, which was similar to a previous study conducted in Addis Ababa (90.8%) (5). The proportion of persons with a successful treatment outcome was lower among underweight patients compared with normal or overweight patients (85.3% versus 93.4%). The odds of a successful treatment outcomes were 2.15 times higher in normal or overweight TB patients versus those that were underweight at baseline. These results are consistent with a South Korean study that found that normal or overweight patients had 3 times higher odds of successful treatment outcomes versus those who were underweight [33]. Additionally, among patients with AFB smear-positive PTB, 7.5% of underweight patients remained smear-positive after 6 months of anti-TB treatment, whereas no treatment failure was found among normal or overweight patients.

Being normal or overweight after 2 months of treatment was also significantly associated with greater odds of successful treatment versus underweight patients (AOR = 3.55; 95% CI: 1.29, 9.73). This is consistent with an Indian study done among adults with TB/HIV co-infection, which found that the risk of developing an unsuccessful TB treatment outcome (failure, death, and default) among underweight patients was higher (AHR = 2.53; 95%CI: 1.87 to 3.42) when compared to normal or overweight patients [24].

Although not statistically significant, a higher proportion of death was observed among underweight patients versus normal or overweight patients (9.24% versus 5.15%). Previous studies in India, Malawi, and Taiwan showed that baseline BMI was associated with significantly increased TB mortality [27–30]. This inconsistency might be due to inadequate sample sizes for death outcomes in the present study. The proportion of underweight patients who died was also comparable with studies from the Dangila and Dessie regions of Ethiopia (7.4% and 9.3%, respectively) [34,35].

The median time to death was lower among normal or overweight patients (36.6 days) as compared to underweight patients (59 days). The lower median time to death among normal or overweight patients compared to underweight patients could be due to comorbidities. Most of the death (52.9% in underweight patients and 85.7% in normal or overweight patients) occurred within 8 weeks after treatment start. A study done in Taiwan indicated that being underweight was associated with early death (within 8 weeks), whereas being normal or overweight was not [30]. Additionally, we observed a slightly higher proportion of the overall mortality occurred within the first 8 weeks of treatment start when compared with the study conducted in Danial, Ethiopia, which reported 56.7% of TB deaths occurring in the first 8 weeks of treatment [34]. This might be due to the differences in sample and study design or uncontrolled co-morbidities.

In this study, we did not find any association between treatment outcome and weight not adjusted for height at any point in time (at baseline, after two and six months) or weight change after two and six months of treatment. This is consistent with a Peru study that found baseline unadjusted weight was not associated with treatment outcomes [36]. Similarly, an Iranian study found weight change at two months of treatment was not associated with treatment outcomes. However, the Iranian study did find weight change after six months of treatment was associated with successful treatment outcomes [37]. Our study is also not consistent with Vietnam and Peru studies which found weight loss and inability to gain was a risk factor for unsuccessful treatment outcomes [31,32]. This inconsistency might be due to differences in sample, study design, setting, and severity of a disease. This study is also not consistent with previous studies conducted in Addis Ababa, Ethiopia that found low baseline unadjusted bodyweight at the start of treatment was associated with unsuccessful treatment outcomes and death [2,22,34,35,38]. This might be due to the difference in study design, sample, and weight categorizing approaches.

Although early nutritional supplementation has previously been associated with body weight gain, body composition [39–41], faster sputum culture conversion, higher cure and treatment completion rate [40,42], a large number (83.8%) of underweight patients claimed that they did not get any food support from any sources. Additionally, more than one-fourth of patients in both underweight and normal or overweight patients claimed that they did not get nutritional counselling. While the rate of food support was very low in the present study (10.8%), it was higher than a previous study conducted in Addis Ababa in which 3.6% of patients received food support [14]. Conducting an assessment of dietary intake, food security, and BMI should be a standard practice during TB treatment, along with dietary counselling [40] and nutritional interventions [43]. Given low BMI was associated with an increased odds of unsuccessful treatment outcomes in the present study, it will be critical to offer improved nutritional support services for these patients in the future.

## Limitations

This study needs to be cautiously interpreted with the limitations listed below. Information on death outcomes was limited to information captured in clinic records and some deaths may

therefore have been missed by this analysis. Most patients were diagnosed with TB disease based on the results of an AFB sputum smear while many were diagnosed clinically. Thus, there is a possibility that some diagnoses of TB disease were incorrect. Additionally, the impact of drug resistance on treatment failure could not be assessed. For persons with TB/HIV co-infection, information on initiation and adherence to antiretroviral therapy was not available.

## Conclusion

Low BMI at the start of treatment was a proxy indicator for worse TB treatment outcomes, while low BMI in the second month was a strong predictor for unsuccessful treatment outcomes. Bodyweight not adjusted for height during treatment and absolute weight change after two and six months of treatment from baseline were not statistically associated with treatment outcomes. One in four patients did not receive nutritional counselling and nearly four in five underweight patients did not receive food support from any sources. We recommend strengthening nutritional assessment, counselling and interventions as a standard practice for TB patients.

## Supporting information

**S1 Checklist. The RECORD statement—Checklist of items, extended from the STROBE statement, that should be reported in observational studies using routinely collected health data.**
(DOCX)

**S1 Fig. Predicted probability of treatment successes by BMI at the start of treatment and sex among adult TB patients in public health centers of Addis Ababa, Ethiopia, 2019.**
(DOCX)

**S2 Fig. Predicted probability of treatment successes by BMI at second months of treatment and sex among adult TB patients in public health centers of Addis Ababa, Ethiopia, 2019.**
(DOCX)

**S3 Fig. Predicted probability of treatment successes by BMI at the start of treatment and age among adult TB patients in public health centers of Addis Ababa, Ethiopia, 2019.**
(DOCX)

**S4 Fig. Predicted probability of treatment successes by BMI at the second month of treatment and age among adult TB patients in public health centers of Addis Ababa, Ethiopia, 2019.**
(DOCX)

**S1 Table. Univariable analysis of nutritional counselling and support association with change in body weight and BMI at the second months of treatment among adult TB patients in public health center of Addis Ababa, Ethiopia, 2019.**
(DOCX)

**S2 Table. Univariable and multivariable analysis of BMI at the start of treatment association with treatment outcomes among adult TB patients in public health center of Addis Ababa, Ethiopia, 2019.**
(DOCX)

**S3 Table. Univariable and multivariable analysis of BMI in the second month of treatment association with treatment outcomes among adult TB patients in public health center of**

**Addis Ababa, Ethiopia, 2019.**
(DOCX)

**S1 File. English version questionnaire.**
(PDF)

**S2 File. Amharic version questionnaire.**
(PDF)

## Acknowledgments

We would like to thank Ms Hannah Nicol for her assistance with editing the manuscript.

## Author Contributions

**Conceptualization:** Zekariyas Sahile.

**Data curation:** Zekariyas Sahile, Robel Tezera, Damen Haile Mariam, Jeffrey Collins, Jemal Haider Ali.

**Formal analysis:** Zekariyas Sahile, Robel Tezera, Damen Haile Mariam, Jeffrey Collins, Jemal Haider Ali.

**Funding acquisition:** Zekariyas Sahile, Robel Tezera, Damen Haile Mariam, Jemal Haider Ali.

**Investigation:** Zekariyas Sahile, Robel Tezera, Damen Haile Mariam, Jeffrey Collins, Jemal Haider Ali.

**Methodology:** Zekariyas Sahile, Robel Tezera, Damen Haile Mariam, Jeffrey Collins, Jemal Haider Ali.

**Project administration:** Zekariyas Sahile, Robel Tezera.

**Supervision:** Damen Haile Mariam, Jeffrey Collins, Jemal Haider Ali.

**Writing – original draft:** Zekariyas Sahile, Robel Tezera, Damen Haile Mariam, Jeffrey Collins, Jemal Haider Ali.

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
