## [Decision Letter · Decision Letter 0]

21 Oct 2020

PONE-D-20-29191

The association between nutritional status and TB treatment outcomes in Addis Ababa, Ethiopia: An ambi-directional cohort study

PLOS ONE

Dear Dr. Sahile,

Thank you for submitting your manuscript to PLOS ONE. After careful consideration, we feel that it has merit but does not fully meet PLOS ONE’s publication criteria as it currently stands. Therefore, we invite you to submit a revised version of the manuscript that addresses the points raised during the review process.

Please submit your revised manuscript. If you will need more time than this to complete your revisions, please reply to this message or contact the journal office at plosone@plos.org. Please include the following items when submitting your revised manuscript:

We look forward to receiving your revised manuscript.

Kind regards,

Frederick Quinn

Academic Editor

PLOS ONE

Journal Requirements:

2. Please clearly state all the variables you included in the multivariable regression model.

3. In statistical methods, please clarify whether you corrected for multiple comparisons.

4. Please include additional information regarding the survey or questionnaire used in the study and ensure that you have provided sufficient details that others could replicate the analyses. For instance, if you developed a questionnaire as part of this study and it is not under a copyright more restrictive than CC-BY, please include a copy, in both the original language and English, as Supporting Information, or include a citation if it has been published previously.

5. As part of your revision, please complete and submit a copy of the RECORD checklist, a document that aims to improve reporting and reproducibility of observational studies that use routinely-collected data for purposes of post-publication data analysis and reproducibility: (http://record-statement.org). Please include your completed checklist as a Supporting Information file. Note that if your paper is accepted for publication, this checklist will be published as part of your article.

6.We suggest you thoroughly copyedit your manuscript for language usage, spelling, and grammar. If you do not know anyone who can help you do this, you may wish to consider employing a professional scientific editing service.  

7.We note that you have indicated that data from this study are available upon request. PLOS only allows data to be available upon request if there are legal or ethical restrictions on sharing data publicly. For information on unacceptable data access restrictions, please see http://journals.plos.org/plosone/s/data-availability#loc-unacceptable-data-access-restrictions.

8.Thank you for stating the following in the Acknowledgments Section of your manuscript:

[This work was supported in part by the NIH/Fogarty International Center Global Infectious Disease

341 grant D43TW009127.]

 [This research work was funded by Addis Ababa University thematic research project fund.  ]

9. Please ensure that you refer to Figure 3 in your text as, if accepted, production will need this reference to link the reader to the figure.

10. Please include captions for your Supporting Information files at the end of your manuscript, and update any in-text citations to match accordingly. Please see our Supporting Information guidelines for more information: http://journals.plos.org/plosone/s/supporting-information.

Reviewers' comments:

Reviewer's Responses to Questions

**Comments to the Author**

1. Is the manuscript technically sound, and do the data support the conclusions?

Reviewer #1: Partly

Reviewer #2: Partly

2. Has the statistical analysis been performed appropriately and rigorously? 

Reviewer #1: I Don't Know

Reviewer #2: Yes

3. Have the authors made all data underlying the findings in their manuscript fully available?

Reviewer #1: Yes

Reviewer #2: Yes

4. Is the manuscript presented in an intelligible fashion and written in standard English?

Reviewer #1: Yes

Reviewer #2: Yes

5. Review Comments to the Author

Reviewer #1: Sahile Z and co-workers performed a retrospective and prospective cohort study of 456 TB patients in Addis Ababa, Ethiopia, comparing the TB treatment outcomes in persons with low BMI (<18.5 kg/m2) vs. those with BMI ≥ 18.5 kg/m2 at treatment initiation and after 2 months of treatment. Treatment outcomes were classified as “successful” – based on 1) negative AFB sputum smear at end of treatment + at least one other previous test or 2) completed treatment with resolution of symptoms. Unsuccessful outcome was defined as those with persistently positive AFB sputum smear at 5 months or later of TB treatment, died, or defaulted from treatment.

Comments:

1. How was the diagnosis of TB (pulmonary or extrapulmonary) made? Based on their aforementioned definition for “successful” and “unsuccessful” outcomes, it seems as if only AFB smear was done. If so, how was the diagnosis made for the smear-negative TB (which made up of ~26-27% of the TB subjects) and of the extrapulmonary TB (14-37% of the subjects)? How do they know that some of the acid-fast bacteria may be non-tuberculous mycobacteria. I think this should at least be mentioned as a limitation of the study.

2. The authors referred to the low BMI group as “exposed” and the normal/high BMI group as “unexposed.” What is the reasoning behind this? This seems unnecessary and actually makes the manuscript more difficult to understand. Indeed, in the last paragraph of the Results and first paragraph of the Discussion, they used the terms underweight and normal/overweight, which is less ambiguous than re-naming the groups to terms (exposed and unexposed) that are cryptic. Alternative, they could refer to the groups as low BMI or normal/high BMI rather than exposed and unexposed.

3. The comparisons in Table 1 between the underweight and normal/overweight patients need P values to determine significant differences (or lack thereof). While relatively smaller in numbers, the underweight group had higher number of prior relapsed TB and greater number who were infected with HIV, which could be confounders for worse treatment outcomes. We also know that failed prior therapy (relapsed) is likely associated with worse prognosis as it is a substrate for resistance to greater number of anti-TB drugs. Also, do you have data on number of prior treatments in the “relapsed” groups?

4. Similarly, the comparisons in Table 2 between the underweight and normal/overweight patients need P values to determine significant differences (or lack thereof). The number of HIV positive cases was ~1.7X greater in the underweight group vs the normal/overweight group and certainly could impact outcome. What % of those who are HIV positive were receiving anti-retroviral therapy as this may obviously impact outcome in several ways including improving host immunity and adverse interactions of anti-HIV drugs with anti-TB drugs.

5. In the abstract, the first sentence of Results section contradicts the first sentence of the Conclusion section. That is, I think the first sentence of the Conclusion should be “….underweight at baseline and after 2 months of treatment was a predictor for treatment failure” (rather than treatment success as it is written).

6. Are there data on the number of TB patients in each of the successful and unsuccessful treatment outcomes with MTB isolates that are rifampin resistant, MDR-TB, or XDR-TB? It seems likely that high levels of drug resistance may actually play a bigger role on whether the outcome is successful or unsuccessful. Along that line, were any patients diagnosed with Cepheid GeneXpert or another nucleic acid amplification test, which may confirm or refute the diagnosis of MDR-TB and thus one can compare patients with drug-susceptible TB with each other and those with MDR-TB with each other.

7. What was the general treatment regimen used and was it different between the underweight vs the normal/overweight groups? Those who were underweight were more likely to be smear-positive…was cavitary disease more common in this group since it may be a marker for more severe disease as well as greater bacterial burden.

8. In line 206, they noted “Among 106 pulmonary TB patients in the exposed group…” I am unclear where this “106” number is derived from. It is not in Table 2 I think.

9. In Table 3, since the UAOR for the weight change for 2 and 6 months after treatment, shouldn’t you have the data for the mean weight changes for the successful and unsuccessful treatment groups for these times after treatment was started? I think it would be important to show to the readers what directionality (weight gain, weight loss) the weight change was going.

10. In the first paragraph of the Discussion, the text goes back and forth unsuccessful and successful outcome. I think it would be less confusing to use one or the other whenever possible and not a combination of both. For example, one could say greater successful outcome vs. lower successful outcome, rather than greater successful outcome vs. higher unsuccessful outcome.

11. In the fifth paragraph of the Discussion, they noted “…we did not find any association between treatment outcome and weight at any time point….” which I do not understand. First of all, they examined BMI and not weight. Second, I thought they DID find an association between low BMI at baseline and at 2 months and lower successful outcomes. They also noted that “This is consistent with a Peru study that found baseline weight was not associated with treatment outcomes.” I don’t understand how this is consistent with the Peru study since low baseline BMI in their study was associated with lower treatment success; i.e., while I acknowledge the Peru study examined weight and the authors here examined BMI, it seems to me that their data, if anything, is NOT consistent with the Peru study.

12. In their Conclusion paragraph, the first sentence is ambiguous and needs more description of directionality; e.g., I think it is better phrased as “Low BMI at start of treatment was a proxy indicator for worse TB treatment outcomes while low BMI after 2 months of treatment was a strong predictor for unsuccessful treatment outcome.

Reviewer #2: • Sample size was based on reference 24 . It has been stated hat it is based on Indian study ‘Kindly check the reference

• Patients in the exposed group have larger number of smear positive cases . Has the correlation with outcomes seen in relation with smear positivity. Has it been considered in adjusted odds ratio.

• Death has been reported to occur in lesser days in unexposed group as compared to exposed group. What is the likely cause . Kindly discuss.

• Kindly check the table 2 . Kindly check outcome numbers for BMI at 2 months .

• Nutritional status has been compared with treatment outcome in various studies. What is the novelty of this study. While BMI is significantly associated with outcome, weight is not associated . kindly explain.

• What was the change in weight seen in 2 months and 6 months. kindly mention the numbers.

• Grammatical errors needs to be corrected.

6. PLOS authors have the option to publish the peer review history of their article (what does this mean?). If published, this will include your full peer review and any attached files.

Reviewer #1: **Yes: **Edward Chan

Reviewer #2: No

---

## [Author Response · Author response to Decision Letter 0]

17 Dec 2020

Dear Reviewers 

We would like to thank you for the comments and suggestions. We read all the comments one by one and prepared a new version manuscript. All responses to comments are attached to a document file named as "Response to Reviewers". Thanks.

---

## [Decision Letter · Decision Letter 1]

28 Jan 2021

PONE-D-20-29191R1

Nutritional status and TB treatment outcomes in Addis Ababa, Ethiopia: An ambi-directional cohort study

PLOS ONE

Dear Dr. Sahile,

Thank you for submitting your manuscript to PLOS ONE. After careful consideration, we feel that it has merit but does not fully meet PLOS ONE’s publication criteria as it currently stands. Therefore, we invite you to submit a revised version of the manuscript that addresses the points raised during the review process.

Please submit your revised manuscript. If you will need significantly more time to complete your revisions, please reply to this message or contact the journal office at plosone@plos.org. Please include the following items when submitting your revised manuscript:

We look forward to receiving your revised manuscript.

Kind regards,

Frederick Quinn

Academic Editor

PLOS ONE

Reviewers' comments:

Reviewer's Responses to Questions

**Comments to the Author**

1. If the authors have adequately addressed your comments raised in a previous round of review and you feel that this manuscript is now acceptable for publication, you may indicate that here to bypass the “Comments to the Author” section, enter your conflict of interest statement in the “Confidential to Editor” section, and submit your "Accept" recommendation.

Reviewer #1: All comments have been addressed

Reviewer #2: (No Response)

2. Is the manuscript technically sound, and do the data support the conclusions?

Reviewer #1: Yes

Reviewer #2: Yes

3. Has the statistical analysis been performed appropriately and rigorously? 

Reviewer #1: Yes

Reviewer #2: Yes

4. Have the authors made all data underlying the findings in their manuscript fully available?

Reviewer #1: Yes

Reviewer #2: Yes

5. Is the manuscript presented in an intelligible fashion and written in standard English?

Reviewer #1: Yes

Reviewer #2: Yes

6. Review Comments to the Author

Reviewer #1: The authors have done a nice job in addressing my previous comments. This is an important study to be added to risk factors for poor TB treatment outcomes.

Reviewer #2: Kindly mention the details of nutritional counselling and nutrition support and how were they provided

Kindly mention relation of nutritional support and nutritional counselling to BMI at 2 month and weight change

7. PLOS authors have the option to publish the peer review history of their article (what does this mean?). If published, this will include your full peer review and any attached files.

Reviewer #1: **Yes: **Edward D. Chan

Reviewer #2: No

---

## [Author Response · Author response to Decision Letter 1]

8 Feb 2021

Dear Reviwer 

We would like to thank you for your comments and suggestions you are given for the above manuscript. We have considered your comments and revised the manuscript. Thank you.

---

## [Decision Letter · Decision Letter 2]

17 Feb 2021

Nutritional status and TB treatment outcomes in Addis Ababa, Ethiopia: An ambi-directional cohort study

PONE-D-20-29191R2

Dear Dr. Sahile,

We’re pleased to inform you that your manuscript has been judged scientifically suitable for publication and will be formally accepted for publication once it meets all outstanding technical requirements.

Kind regards,

Frederick Quinn

Academic Editor

PLOS ONE

Additional Editor Comments (optional):

Reviewers' comments:

Reviewer's Responses to Questions

**Comments to the Author**

1. If the authors have adequately addressed your comments raised in a previous round of review and you feel that this manuscript is now acceptable for publication, you may indicate that here to bypass the “Comments to the Author” section, enter your conflict of interest statement in the “Confidential to Editor” section, and submit your "Accept" recommendation.

Reviewer #1: All comments have been addressed

2. Is the manuscript technically sound, and do the data support the conclusions?

Reviewer #1: Yes

3. Has the statistical analysis been performed appropriately and rigorously? 

Reviewer #1: Yes

4. Have the authors made all data underlying the findings in their manuscript fully available?

Reviewer #1: Yes

5. Is the manuscript presented in an intelligible fashion and written in standard English?

Reviewer #1: Yes

6. Review Comments to the Author

Reviewer #1: The authors have written a very nice paper which will help clinicians care for patients with tuberculosis.

7. PLOS authors have the option to publish the peer review history of their article (what does this mean?). If published, this will include your full peer review and any attached files.

Reviewer #1: No

---

## [Editor Report · Acceptance letter]

19 Feb 2021

PONE-D-20-29191R2 

Nutritional status and TB treatment outcomes in Addis Ababa, Ethiopia: An ambi-directional cohort study 

Dear Dr. Sahile:

I'm pleased to inform you that your manuscript has been deemed suitable for publication in PLOS ONE. Congratulations! Your manuscript is now with our production department. 

Kind regards, 

on behalf of

Dr. Frederick Quinn 

Academic Editor

PLOS ONE